# An Attempt at a Theoretical Explanation of Violent Islamist Radicalization in Spain

Sergio García Magariño [1,*] and María Jiménez-Ramos [2]

1   I-Communitas, Public University of Navarre, 31006 Pamplona, Spain
2   Journalistic Projects, University of Navarra, 31009 Pamplona, Spain; mjimenezr@unav.es
*   Correspondence: sergio.garciam@unavarra.es

**Abstract:** This paper is the first in a series of papers that aim to address Islamist violent radicalization from different angles: the nature of violent radicalization in the context of Spain; a comparison between European, North American, and Indian violent radicalization; the need to refine territorial radicalization indexes within the context of preventing violent radicalization and the relation between Islamist violent radicalization; and other forms of violent radicalization in Europe. This series of articles builds upon the general theoretical framework established by the author in two previous works. These works are framed under the known conception of three layers of micro, meso, and macro factors contributing to violent radicalization processes. The paper starts by defining Islamist violent radicalization. Then, it explores different theoretical explanations, and finally, it proposes an explanatory hypothesis that is tested against, on the one hand, data proceeding from different institutional sources in Spain and, on the other, some initial conversations to Spanish security officials and people who were radicalized in the past and regretted it or lived very close to others that did it. In further articles, these preliminary conversations will become life stories and in-depth interviews.

**Keywords:** jihadism; radicalization; moral structure; extremism; social control

## 1. Introduction

Research on violent radicalization is as necessary as it is complex. It is necessary because peace and social cohesion are the basis for social progress, so preventing any kind of political violence is crucial. However, it is also complex because rigorous study of violent radicalization requires conceptual clarity, theoretical depth, mastering innovative methodologies and empirical grounds. This paper aspires to provisionally explore some apparently promising lines of inquire into the four dimensions indicated before: conceptualization, theoretical explanation, searching for innovative methodologies and a quest for empirical evidence.

Although there is no consensus on its meaning, radicalization could be described as a complex nonlinear process where structural, cognitive, and attitudinal factors lead an individual or collective to question and reject the norms of the society or the group to which they belong. In other words, it could be said that this process results in the person or group being placed on the margins of the society and moving away from "normality".

In addition to being individual and collective, radicalization could be positive or negative (García-Magariño 2019a). The positive radicalization is related to a critique of the status quo and advocates for, through nonviolent means, the creation of a more just, sustainable, and peaceful order. Historical figures such as Martin Luther King, Nelson Mandela, or Mahatma Gandhi, who today are considered to have led major social transformations, were described as radicals in their time. This possible positive meaning of radicalization is similar to the conceptualization of Mannheim (1987), refined by Ricoeur (1989), regarding the social role of the ideology—as an alternative vision that questions the prevalent order and reveals its shortcomings. Positive radicalization would have, in this way, a social function comparable to that of utopia.

On the contrary, regarding its negative aspect, radicalization justifies and legitimizes the use of different types of physical and verbal violence. In fact, the concept of terrorism includes violent acts that are carried out systematically to achieve a goal, be they individual or collective, that are targeted against a state or civilians, and that exploit terror intentionally (Calera 2002). Nevertheless, not all those who are radicals or undergoing the process of violent radicalization end up committing violent acts. That is, not every radical is a terrorist, but every terrorist is a radical person (CPRLV 2016). Typifying radicalization or elaborating a detailed taxonomy of the same exceeds the scope of this document. However, from the previous introduction, several categories of radicalization can be identified: (a) positive or negative; (b) individual or collective. Several axes could be introduced such as motivations—political–ideological, religious, or criminal—or the intensity and the type of violence used—high, low, verbal, physical.

At this point, it may be necessary to observe that although negative radicalization is often associated with individuals, it may also affect entire societies. As radicalization always occurs when a broader collective moves away from the center toward the periphery of *the normal*, the reference frame may be that of human rights and the basic norms of peaceful coexistence. Thus, when a society begins to normalize extreme approaches close to violence and begins to develop an overly polarized culture, it can also be said to have started a radicalization process directed towards the margins of the minimum ethical and moral framework accepted by the international community, a framework that is expressed mainly in the language of human rights.

According to the goals they follow, radical terrorist groups that advocate the use of violence can also be grouped in different categories: nationalist groups or separatists seeking independence for a territory, such as the terrorist band Euskadi Ta Askatasuna (ETA) in Spain or the Irish Republican Army (IRA); groups of the extreme right that defend the superiority of the white race and the rejection of immigration, such as the Ku Klux Klan in the United States or Golden Dawn in Greece; groups of extreme left that seek social justice, the redistribution of wealth and which are opposed to capitalism, such as the Revolutionary Armed Forces of Colombia (FARC) in Colombia or the Red Brigades in Italy; groups with religious motivation which fight for the implementation of a narrow interpretation of the religious writings and the conversion of nonbelievers, such as the Army of God in the United States, the Islamic State of Iraq and Siria (ISIS) or Al Qaeda; and lastly, groups focusing mainly on a single social issue such as the environment, animal rights or abortion (Sant 2019).

It seems that radicalization becomes more dangerous when religion is involved (García-Magariño 2015). The reason is that this entails a deep motivational force that makes the person willing to sacrifice themself for later generations, renounce their well-being, and take any action that they interpret as the will of God. Therefore, one of the assumptions of our ongoing research is that, when radicalization relates to religious motivation, it is necessary to pay attention to religious logic to find viable solutions that do not produce unexpected side effects that aggravate the problem in the long term. This is a classic study topic in sociology that has several implications, one of which has already been examined in depth by Durkheim in *El Suicidio*, with such singularity that its significance is practically boundless.

The kind of radicalization explored here is, more particularly, the Islamic violent radicalization. In an attempt to synthesize, it could be said that while Islam is a religion which is more than 1400 years old, Islamism is a modern political current that advocates applying the social, political, and economic dimensions of Islam to the collective organization of the society. Within the Islamism currents, there are different versions, some of them advocating for democracy and others for the establishment of the Islamic States where Islamic law, called sharia, is applied. Among those who advocate the establishment of sharia law, some groups opt for the peaceful way of doing things, which means winning elections and then starting from there to make changes as its majority grows; others choose violence and consider revolutionary action to be the axis of their transformation strategy

(García-Magariño 2016). The latter can be divided into groups with national aspirations, which adopt nationalist approaches and internationalist groups which seek to extend the *umma* (the religious–political Islamic community) to a constantly expanding transnational territory within which sharia rules. Modern Salafism is a rigorous Islamist current, confined to Saudi Arabia decades ago but prevalent today in the world owing to its growth linked to oil money. All Salafist groups do not advocate violence but most Islamist groups that champion revolutionary and violent action emerge from Salafism and draw inspiration from different versions of it. Finally, the notion of jihad is often embraced with a double meaning among Muslims: an individual effort to better oneself and to improve individually, and defensive warfare. Some Islamist currents also refer to jihad as a preventive defensive war in advance, although the Al-Qaeda and Daesh ideologues apply the jihad to extend their territorial model or achieve political goals.

For all these reasons, violent Islamist radicalization is often called violent Salafist radicalization, Salafist–jihadist radicalization, or jihadist radicalization. Beyond its name, it is the object of study of this research, particularly the Spanish case. The expression 'violent Islamist radicalization' will be used in this article because it is probably the most accurate and appropriate, although Salafist–jihadist radicalization or just jihadist radicalization are both widely used.

## 2. Theoretical Explanations of Violent Radicalization

Theoretical efforts to explain violent radicalization from the causal and universal standpoints, as well as single-factor explanations, have been unsuccessful. Although some theories gained popularity and generated enthusiasm, they have not been verified with extensive empirical evidence from different countries. This circumstance has led researchers to conclude that there is no single profile. Some of them have been more cautious in proclaiming that the profile has not yet been discovered (López Melero 2017). The following paragraphs will discuss some of the major theories and explanations that have been used to explain such an elusive phenomenon. Finally, we will suggest our explanation regarding the reasons why universal concepts have not been discovered. We will focus on the diversity of the phenomenon, the difficulty of gaining first-hand access to those who have become radicalized and acted violently, the idiosyncrasy of each country and region, and the need for hypotheses with considerable heuristic potential.

Regarding the initial explanations for violent Islamist radicalization, it was held that terrorists came from poor and marginalized environments, mainly from the Arab world. Therefore, poverty, oppression, and exclusion were used as explanatory vectors. The September 11 attacks in the United States as well as the people who joined Al-Qaeda in the United Kingdom revealed that rich youths, including others from the middle class with higher education, were the main architects/authors of the attacks (McCauley and Moskalenko 2017). This explanation which, in addition, excluded the religious fact, resulted in many initial studies going to the other extreme of taking only religion as a causal explanation. That pendulum, that tension between exclusively secular, social, economic, and identity explanations, which still exists, impedes the understanding and explanation of such a complex phenomenon which cannot be unraveled by a single variable.

When this fact was established, a general framework of factors was developed, to which reference was made in the introduction, that leads to radicalization: individual motivations—of the rational, identity, normative and emotional kind; meso-sociological contextual factors—the group, known offenders, the radicalization agent; and macro-structural—large-scale conflicts, the media, armed forces and police action, the existence of terrorist organizations. This pattern is a good matrix to approach radicalization, but at the same time, it is so broad that it does not explain the process precisely.

After this first stage, the explanation had to become more sophisticated. Thus, two types of explanatory theories emerged: the staggered and progressive ones, which try to explain the individual trajectory of violent radicalization; and the pyramid ones, which focus on collective dynamics. As for the sequential theories,

they state that the person goes through critical stages. Depending on the author, the number and nature of the stages differ. However, all of them start from the same premise: the person moves towards progressive levels of radicalization, from normal to violent behavior. Some stages would be exposure to radical ideas, be they political, social or religious—that is the reason why, for example, in Spain, the consumption of jihadist content on the internet carries a sentence—the assumption of a new identity, association with radical groups and justification of violence, planning of small violent actions and their subsequent execution, etc. As a result, individuals reject diversity, tolerance, and freedom of choice, and they legitimize breaking the rule of law and using violence towards property and people (Schmid 2013). Sequential models change, but they are all based on the same logic: anybody can radicalize when certain factors are present, and in addition, the process is regressive and goes from radical thinking to violent action. (Moghaddam 2005; Horgan 2005; Reinares et al. 2019)

Pyramid theories follow a similar pattern but involve social dynamics. The first level of the pyramid is composed of politically neutral people. Violent radicalization would depend on a broad base of people, on a second level, who sympathize with the violent Islamist cause. Then we would have another level of people who justify the use of violence. Finally, there would be the level of those who commit to the cause and join the terrorist group or cell (Leuprecht et al. 2010).

The fact that only a very small number of those who justify violence take the effective step forward led to the questioning of these explanations and to the formulation of a new explanatory model, that of the double pyramid. The double pyramid means in essence that cognitive radicalization (a pyramid) and behavioral radicalization (the other pyramid) are not connected. That is, it is required to explain cognitive radicalization, on one hand, and behavioral radicalization on the other. From another angle, it can be said that this new proposal dissociates thought and action. Authors like Oliver Roy, without being the architects of this theory, have given empirical support to this explanation by showing how in France, for example, many of the terrorists were neither especially religious nor had they experienced a gradual process of radicalization, but rather an abrupt conversion (2017). This explanation seeks other factors to explain violent behavior such as the previous history with violence, strong uprooting, traumatic experiences, or strong networks of identity and recruitment. What is sometimes posed is that those who join global jihad or Islamist terrorist groups and cells were previously radicalized in search of a cause that could channel their desire to manifest that unease or euphoria: it could have been both revolutionary communism and violent anarchism (McCauley and Moskalenko 2017).

The other side of the coin concerning this hypothesis consists of identifying the factors that, in similar contexts, can act as inhibitors of the use of violence. Alonso has identified some of them: (i) Existence and recognition of political opportunities and alternatives to violence; (ii) Violent means deemed unjustifiable despite the agreement with some of the group's ends; (iii) Belief in the inefficacy of violence and its counter-productive effects; (iv) A more tolerant and less fundamentalist and fanatical approach to ideology; (v) Ambivalence towards the terrorist group, fear; (vi) Lack of prestige in violence, morality; (vii) Rejection of the terrorist group's criminal activities and justifications; viii) Lack of precipitant events and connectedness at a vulnerable age; (ix) An alternative socialization away from the subculture of violence; (x) Kinship and social ties, personal responsibility to family members, professional opportunities; (xi) Empathy towards victims of terrorism (Alonso 2021).

Despite the evidence, accepting that thinking, attitudes, and action are not connected is quite complicated. In the following section, an alternative hypothesis will be offered to resolve this apparent dilemma; before doing that, this point will be brought to a close, as previously mentioned by retrieving, on the one hand, some elements of consensus beyond the universal explanations and outlining, on the other hand, the reasons why it becomes challenging to find a more general theoretical explanation for this phenomenon.

### 2.1. The Terrorist Profiles

As to finding consensus, profiling is perhaps the aspect that generates the greatest consensus because it simply involves drawing up an average for categories such as gender, age, national origin, nationality, educational level, socioeconomic status, place of residence or radicalization time concerning all those accused of, for example, belonging to armed groups or glorification of terrorism. It is necessary to differentiate between those who have engaged in attacks or tried to do so in their country of residence, and those who have traveled to Syria or Iraq to join Daesh. Data may vary slightly between European countries and even more when Saudi Arabia, Morocco, Pakistan, or Argelia are taken as reference. In Spain, the Elcano Royal Institute conducts detailed profiles based on police and judicial data every two years (Figure 1) (Reinares and García-Calvo 2013). It is worth mentioning that the profile in Spain—among more than three hundred cases—is comprised of the following: (a) men—in recent years there have been a few women; (b) unconverted first and second generation Muslims—however, converts are beginning to emerge and the percentage of converts who opt for radical ways is higher than the general percentage of Muslims, so it is considered a growing trend; (c) youth; (d) nationals, but pertaining to families from Arab countries; (e) members of the lower middle class—this differs from what happens in other countries; (f) previously radicalized relatives or friends; (g) dissidents of their parents religion who adopt jihadist–Salafism in an almost abrupt manner. They radicalize from top to bottom by means of an agent of radicalization. In Spain, there are no people who self-radicalize on the internet, although the internet is increasingly used as a support, and they radicalize in Catalonia, Ceuta, Melilla and to a lesser degree in Madrid and Valencia, to mention some data (Reinares et al. 2019).

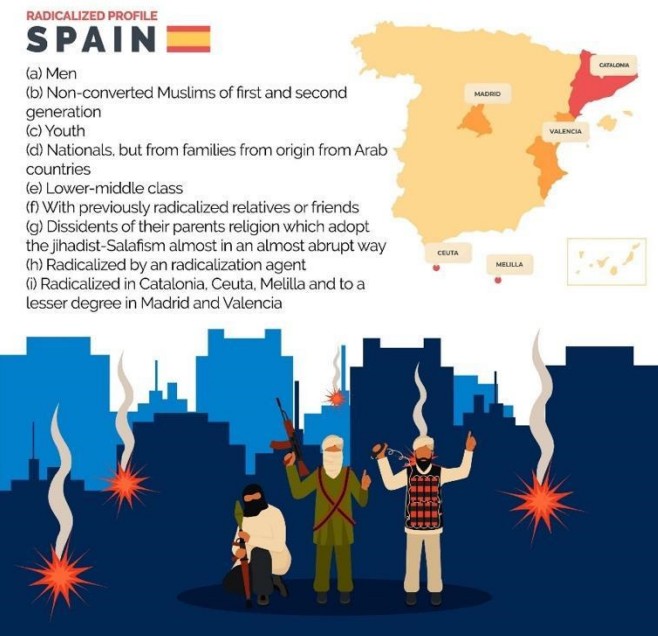

**Figure 1.** Profile of radicalized people in Spain. Created by Vahid Digital Marketing.

This profile, however, does not fully correspond to other countries in Europe. Professor López Melero, who has had access to the penitentiaries in Spain, usually draws up good profiling which includes other variables such as the modus operandi (2017).

### 2.2. The ABC Theoretical Model

Despite the evidence, accepting that thoughts, attitudes, and actions are not connected is complicated. The Attitudes-Behaviors Corrective Model on Violent Extremism (ABC), developed by Horgan—author of the famous book Psychology of Terrorism—aims precisely to fill this gap. The ABC theoretical model aims to correct the explanations that dissociate

thought from action in the process of violent radicalization that leads to terrorism. Indeed, they even argue that it is preferable to abandon the notion of "radicalization" because it overemphasizes cognitive radicalization. They opt instead to talk about the process that triggers violent extremism and terrorism. In essence, the explanatory model posits that the path leading to terrorist action is best understood along two axes, one indicating greater or lesser sympathy for ideologies that advocate violent action as a path to social change, and the other indicating greater or lesser involvement in the practice of such violence. From these axes, they interpret the different modalities of radicalization that may appear. Figure 2 illustrates these vectors and includes some examples of variants of radicalization that are explained in line with the theory.

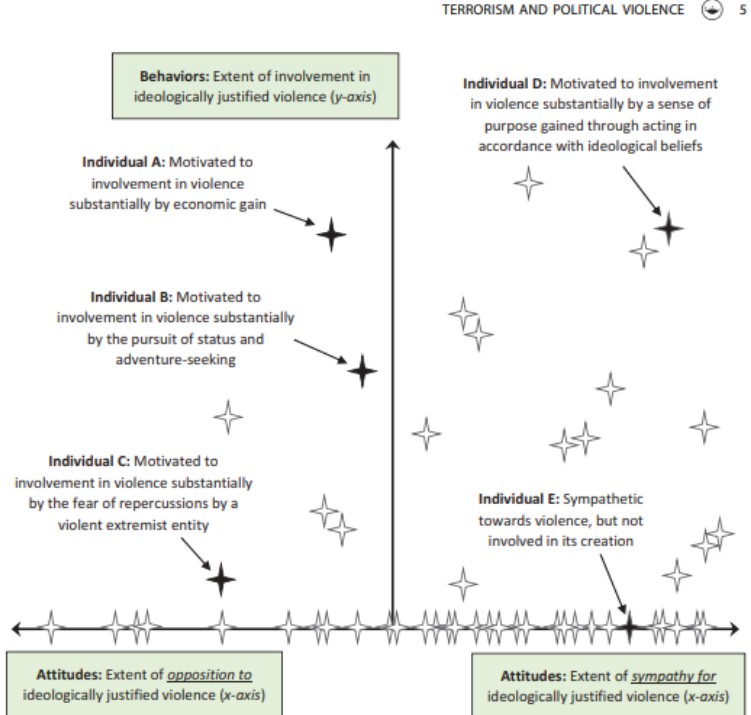

**Figure 2.** The Theoretical Model ABC. Diagram extracted from Khalil et al. (2019) The Attitudes-Behaviors Corrective (ABC) Model of Violent Extremism, Terrorism and Political Violence, DOI:10.1080/09546553.2019.1699793[1].

### 2.3. External Pull Factors and Internal Push Factors

Other perspectives which reach a certain level of consensus are the consideration of the Salafist–jihadist matrix as the breeding ground where radicalization takes place but in combination with other factors that are usually specified in two categories: internal push factors and external pull factors (Table 1) (Mirchandani 2017). Regarding internal factors, there are existential crisis episodes, which are common to almost all the terrorists: real or perceived oppression, real or perceived exclusion, anger and frustration due to unfulfilled expectations—the higher these are, the more difficult they are to satisfy, which connects both poor and unsatisfied middle classes or rich people with a desire for recognition, to mention several apparently opposite profiles—a lack of sense, a search for ties, for belonging and bonding, and a desire for recognition and a better life (Torres-Marín et al. 2017). The pull factors would consist of the appeal of the Salafist–jihadist ideology itself because in a very simple (and reductionistic) manner, it explains Muslim individual and collective marginalization and offers a direct and clear alternative based on the beauty of the group belonged to, fascination for adventure and violence, the possibility of being acknowledged, of having a life which makes sense, in the other life (if you commit suicide) or in the caliphate, etc. If a prior history of violence or criminality is added to that, the process is

accelerated because there are no internalized social control barriers that stop the person from committing antisocial actions (Lemon et al. 2018).

**Table 1.** Factors that favour the radicalization.

| External Pull Factors | Internal Push Factors |
|---|---|
| Episodes of existential crisis | Appeal of the Salafist–jihadist ideology |
| Real or perceived oppression | Beauty of the group belonged to, the cell; |
| Real or perceived exclusion | Fascination for adventure and violence |
| Anger and frustration stemming from unfulfilled expectations | Possibility of being recognized, having a sense-filled life, in the other life (suicides) or in the caliphate, etc. |
| Lack of sense and life-existential meaning | |
| Search for ties, belonging and bonding | |
| Recognition and a better life desire | |

Own elaboration.

The so-called shift to resilience approaches reveals strength factors and individual and collective vulnerability (Stephens et al. 2021). These authors discover, for example, the importance of family, social (integration), and community (religious) roots. Those who have deep roots are less vulnerable (Baobaid and McQuaid 2016). Those individuals with higher levels of religious and scientific knowledge opt for the contextualized versions of Islam, which delegitimize violence.

### 3. The Need for Empirical Studies and Indicators for Studying Radicalization

Regarding the difficulties of finding explanatory theories with universal validity, it should be mentioned that, on one hand, conditions are different in each country. Saudi Arabia is not the same as Spain, France, Nigeria, or Russia. On the other hand, the path of those who decide to commit a terrorist attack in their residence country probably differs from those who decide to go to Syria or Iraq, those who just offer logistic or economic support, or those who engage in swindling. In the same way, the terrorist organization has a structure and differentiated roles. Those in the upper echelons are different from those who are recruited in Afghanistan to transport weapons, those who are sent to ISIS as doctors or engineers, those who operate in rural areas in Mali or the imams, who are the ideologues behind attacks in cities like Barcelona.

In addition, accessing the primary data about those who join the armed conflict is almost impossible, so deciphering complex data to reveal patterns or to rebuild life trajectories, in a bid to find similarities, is a major challenge. In Spain, for example, deep data related to this kind of prisoner are not public, and special permits are required to access them. If the intention is to interview those who have been accused, the situation is still problematic: they are protected by law, and procedures for universities or research centers to receive state permission to enter prison institutions can take two years. Finally, it is vital to mention that, unless the person has repented and wants to collaborate, in Sunni and Shiite Islam, there is the precept of *taqiyya*, the possibility of lying in exceptional cases to protect the community. That means that information obtained through radicalized people may not be reliable, unless they have sincerely repented and want to collaborate. In this sense, one avenue to explore has to do with the family because often the closest relatives of the radicalized person have witnessed the radicalization processes. Their vision, which can be critical or condescending, can, in any case, provide valuable information, which is otherwise inaccessible (Jiménez Ramos 2020).

The phenomenon is complex and multidimensional, so there are multiple possible hypotheses and assumptions, which may lead to a further empirical inquiry, an inquiry that, as has been said, is not simple. It might be the case that no categories with sufficient

heuristic and interpretative power have yet been obtained to approach this social problem with a generalizing explanatory capacity that is effective.

Indicators are vital for the prevention of radicalization. Some indicators portray the stages of the violent local radicalization process and allow for early detection; there are contextual indicators that determine the risk of radicalization and attacks within a territory; there are indicators that assess the impact of programs, and there are indicators that aim to reflect individual and collective resilience to radicalization.

The issue of indicators is closely related to another line of action: the creation of communities that detect these signs, that know how to use these indicators and that, in addition to alarming, can act to reverse the process. These communities not only act as a warning system but also become environments in which resilience is cultivated to resist and overcome the agents and factors that often fuel radicalization. Social workers, school teachers, local police, families, neighborhood associations, health professionals in local centers, and researchers make up such a community, which, moreover, must be geographically manageable, roughly the size of a neighborhood in cities or a village in towns (Green et al. 1998; Leventhal and Brooks-Gunn 2000).

The following are the preliminary results of two ongoing investigations promoted by the University of Kent and the Public University of Navarra, respectively, that aim to study violent radicalization taking the geographic space, specific neighborhoods, as the meso level of analysis. While the material socio-economic profile of radicalized individuals might greatly vary, it is possible that the socio-economic characteristics of the most proximate environment in which radicalized individuals socialize (neighborhoods) present similar characteristics of socio-economic deprivation. In turn, specific types of socio-economic deprivation at a neighborhood level might increase the risk of radicalization by affecting, for example, its citizens' sense of inequality and belonging. In fact, neighborhoods are the most proximate social context in which young adults socialize, make sense of their social world, and learn the rules and regulations necessary to become competent members of society (Leventhal and Brooks-Gunn 2000; Steinberg 1990).

*3.1. Local Socio-Economic Deprivation and Radicalization in England*

Answers as to whether individuals who live in more socio-economically deprived local areas are more likely to be radicalized and turn to violent extremism, and what type of socio-economic deprivation is likely to affect the risk of individual radicalization and why, have so far remained elusive. Scientific knowledge on these questions is particularly salient in England, where the delivery of measures to prevent violent extremism is underpinned by the understanding of the risk of radicalization at a local level (CONTEST 2018). While existing studies on the relationship between inequality/deprivation indicators and radicalization at national levels present contradictory findings and studies on individual characteristics relevant to inequality and cognitive radicalization show that there is not a single profile of 'terrorist' (DARE Project 2018), University of Kent takes a third path. They focus on the socio-economic characteristics of the most salient social context, in which individuals socialize and might radicalize that of English neighborhoods.

They use two proxies to capture levels of radicalization across Local Authority Districts in England. First, they use data on racially and ethnically motivated hate crimes recorded by the English police and published annually by the Home Office (Hate Crime Statistics page on GOV.UK). The logic of using hate crimes as an indicator of the local level of radicalization is based on the assumption that perpetrators of hate crimes tend to commit this type of crime locally. A recent report shows that perpetrators of hate crimes are hard 'strangers' and that hate crime hotspots can be identified over space (Colin et al. 2013). Crucially, they conceptualize hate crimes to be one of the possible outcomes of local processes of radicalization. These types of criminal offenses have in fact a specific racial or religious motivation; they are defined by statute and constitute a set of offenses that are distinct from their nonracially or nonreligiously aggravated equivalents. Racially or religiously aggravated offences are by definition hate crimes. They derive information on

the cumulative number of hate crimes across English Local Authority Districts from 2015 to 2018 from the police recorded crime open data tables published by the Home Office (Figure 3).

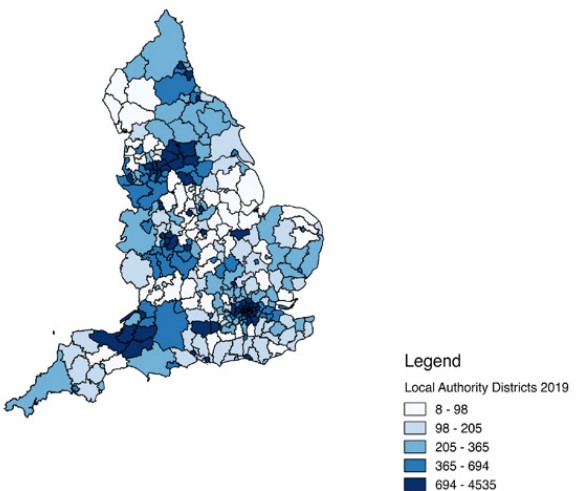

**Figure 3.** Level of radicalization (# of hate crimes 2015–2018) across English Local Authority Districts.

Second, they self-code profiles of perpetrators of terrorist attacks taking place in England between 2015 and 2018 focusing on the location of their radicalization. Profiling terrorist perpetrators with open-source information on news reports is notoriously difficult. They are able to obtain data on the location of radicalization at a Local Authority District Level for 35 perpetrators of terrorist attacks. Information on terrorist attacks taking place in England is initially extracted from the Global Terrorist Database (START). Lexis Nexis and newspaper searches are then used to identify the location of radicalization for each perpetrator (Figure 4). The coding rules are the following: (1) explicit information on where perpetrators appeared to have been permanently residing before the attack. If this information is not available, they look for an indication that the perpetrators were local. They deem perpetrators to be local if there is information showing that (1) the police believed that perpetrators were local, (2) the police call on local communities to come forward for information on perpetrators, (3) there is evidence of local radical groups targeting the same or similar types of targets in the local area, and/or (4) there is evidence of violent attacks against the same or similar types of targets in recent past history reported by the news agency as being systematic.

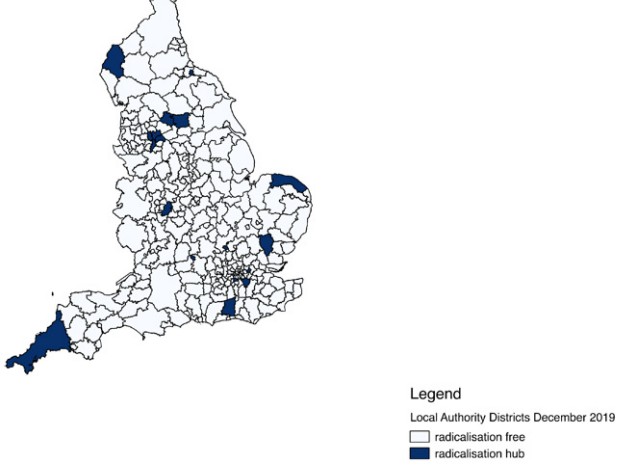

**Figure 4.** Level of radicalization (location of radicalization of perpetrators of terrorist attacks) across English Local Authority Districts.

As independent variables, to capture different types of severe local deprivation in England, they use the disaggregated indicators of the English Indices of Deprivation 2019. The English Indices of Deprivation measure relative levels of deprivation in 32,844 neighborhoods in England. The indicators used to build the 2019 version of the English Indices of Deprivation are based on the latest available data gathered between 2015 and 2016. It is crucial for the analysis because it implies that the measures of types of deprivation precede the processes of radicalization measured by our dependent variables.

The deprivation indicators are described below, and for each of them, the plots show a consistently positive covariance between various aspects of deprivation and our measures of radicalization. This indicates that the higher the level of deprivation within a local authority district, the more manifestation of radicalization one can observe in that district. However, it is important to bear in mind that a complete statistical analysis is needed to confirm the chain of causality between the various indices of deprivation and radicalization at a local district level.

1.  Living Environment Deprivation: it measures the quality of the local environment. The indicator is composed of a measure of 'indoors' living environment measuring the quality of housing and 'outdoors' living environment measuring air quality and road traffic accidents (Figure 5a,b).
2.  Barriers to Housing and Services: it measures the physical and financial accessibility of housing and local services. The indicators fall into two subdomains: 'geographical barriers', which relate to the physical proximity of local services, and 'wider barriers' that include issues relating to access to housing such as affordability (Figure 6a,b).
3.  Health Deprivation: it measures the risk of premature death and the impairment of quality of life through poor physical or mental health. The domain measures morbidity, disability, and premature mortality but not aspects of behavior or environment that may predict future health deprivation (Figure 7a,b).
4.  Employment Deprivation: it measures the proportion of the working-age population in an area involuntarily excluded from the labor market. This includes people who would like to work but cannot do so due to unemployment, sickness or disability, or care responsibilities (Figure 8a,b).
5.  Education, Skills and Training Deprivation: it measures the lack of attainment and skills in the local population. The indicators fall into two subdomains: one is related to children and young people, and one is related to adult skills (Figure 9a,b).
6.  Income Deprivation: it measures the proportion of the population in an area experiencing deprivation relating to low income. The definition of low income used includes both those people who are out-of-work and those who are in work but who have low earnings (Figure 10a,b).

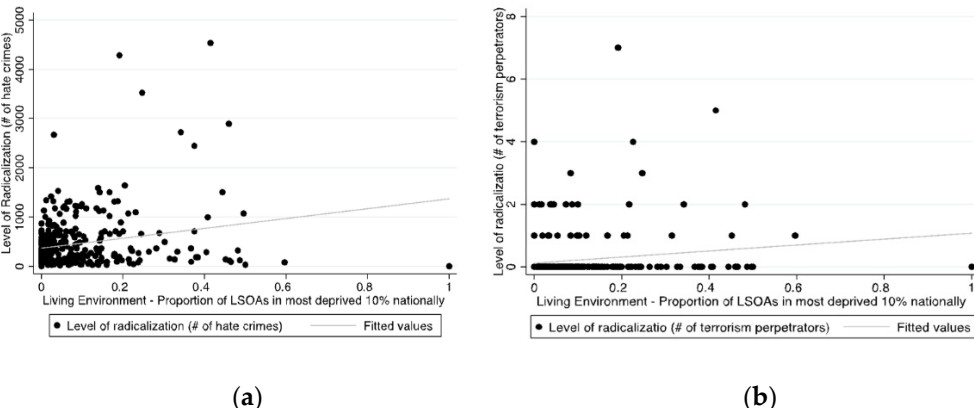

(a)　　　　　　　　　　　　　　　　　　　　　　　(b)

**Figure 5.** (**a**,**b**) Living Environment Deprivation.

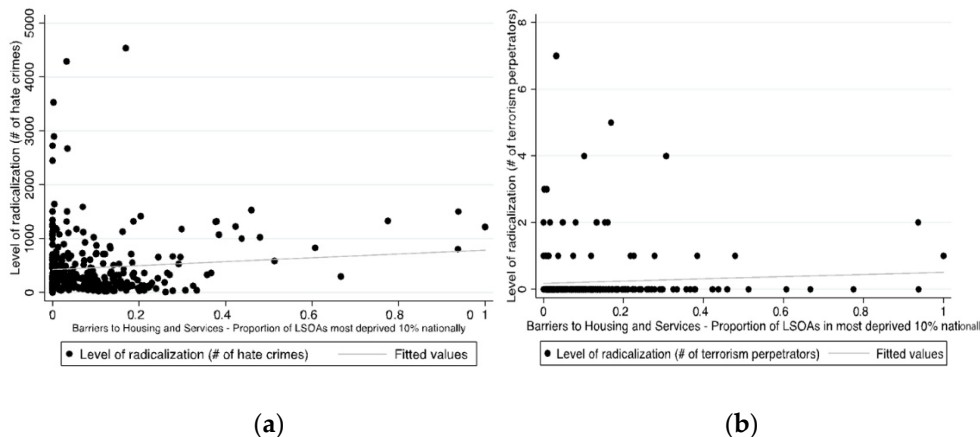

(**a**)                                                                 (**b**)

**Figure 6.** (**a**,**b**) Barriers to Housing and Services.

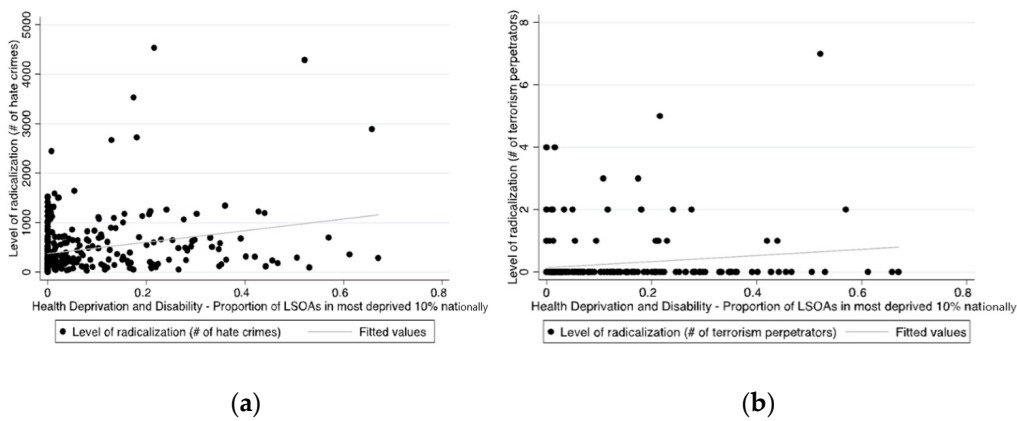

(**a**)                                                                 (**b**)

**Figure 7.** (**a**,**b**) Barriers to Health Deprivation.

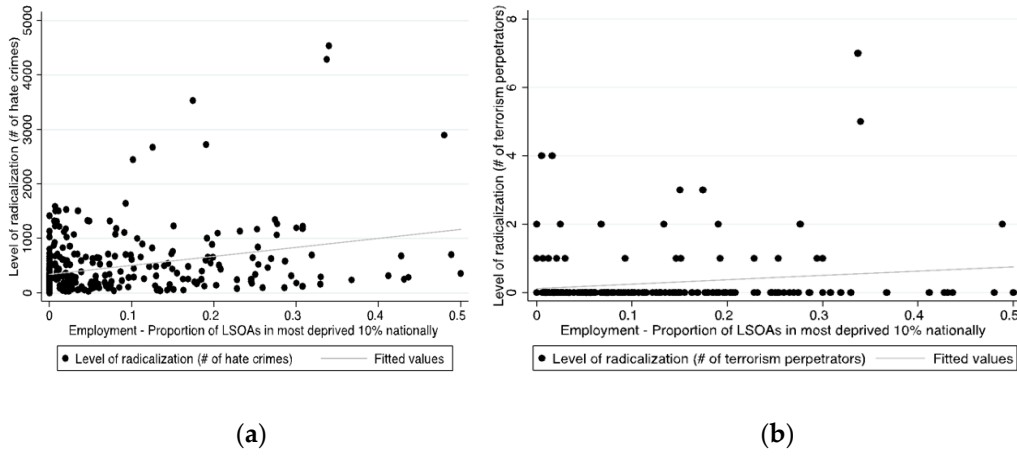

(**a**)                                                                 (**b**)

**Figure 8.** (**a**,**b**) Barriers to Employment Deprivation.

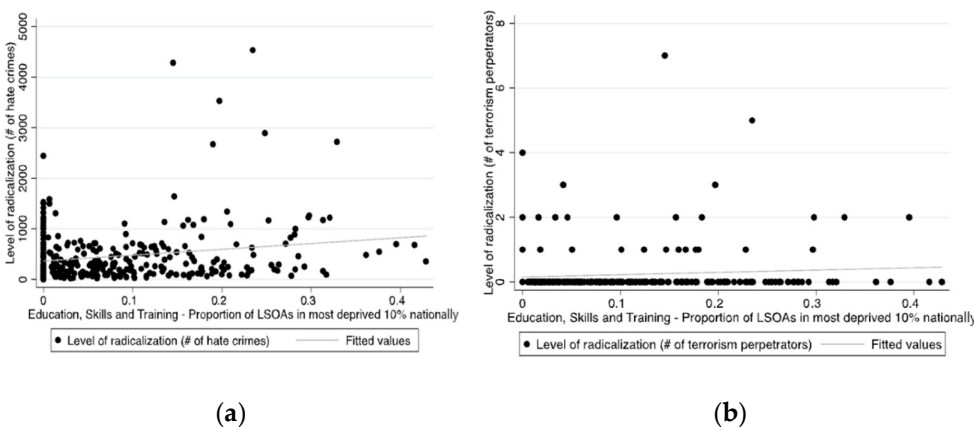

(**a**)

(**b**)

**Figure 9.** (**a**,**b**) Barriers to Education, Skills and Training Deprivation.

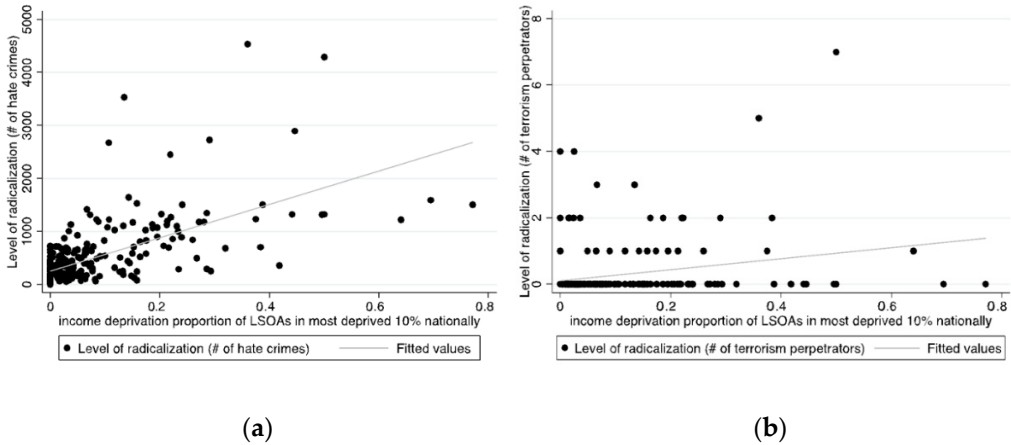

(**a**)

(**b**)

**Figure 10.** (**a**,**b**) Barriers Income Deprivation.

### 3.2. Local Factors That Increase the Risk of Radiation: An Approach

The research project, in which this article is framed, is developing a set of local indicators that could alert to the risk of radicalization. Preliminarily, the author has modeled the primary factors that should be considered in a territory, in the light of empirical data and explanatory theories, to assess the greater or lesser potential for risk of violent radicalization. The parameters are described below.

Before addressing the metrics in a particular way, however, geographic space must be discussed. An indicator of violent radicalization should be the ability to scale from the smallest scope, which would be a district or large neighborhood, to a broader one, that of a province. In addition, it should be projected on maps to facilitate diagnosis and narrow down prevention policies and programs.

As for metrics, it should be borne in mind that the indicators that make up an index must meet three conditions: (a) they measure what they are intended to measure; (b) it is possible to obtain the desired data; (c) they are meant for total violent radicalization. Considering these criteria and the multiple factors described above, the case of jihadism is taken, and the risk-increasing criteria are listed:

1. Existence of a mosque in the neighborhood district with Salafist ideology.
2. In the district, there is a concentration of Muslims living together in the same territory, exceeding the average concentration.
3. The neighborhood district has worse social and economic conditions than the rest of the city.
4. There have been people previously radicalized or accused of terrorism in the neighborhood-district
5. There have been complaints of hate speech against the Muslim population.

6.  Intelligence services have detected agents of radicalization in the area.
7.  There have been previous operations against terrorists in the area
8.  There is a sense of grievance among the Muslim population
9.  Level of affective polarization among the population
10. There are overly homogeneous social groups that identify themselves by nationality and religion
11. Percentage of unaccompanied minors

These metrics are meso-sociological. The indicator, however, considers two additional factors: (a) a social, regional, national, or international event that mobilizes the Muslim population, such as a war, an intifada or an alarming episode of hate crime that goes public; (b) the international advance or retreat of Daesh, Al-Qaeda and their franchises.

Once the scales have been identified, four processes remain: coding, quantification, data collection, and map generation. Coding requires specifying what each of the 11 indicators listed above is intended to assess and detailing how that indicator is to be objectified. Quantification, in turn, requires assigning a score to each metric, depending on its greater or lesser relevance to the process of violent radicalization.

Data collection deserves special attention. Data related to terrorism, especially if they involve issues related to religion and criminal records, are very difficult to obtain because they are often protected by laws that attempt to preserve the integrity of individuals. Moreover, the institutions that handle them are often in partial possession of them. Thus, there is no centralized command that introduces them on the same basis. A promising line of inquiry is therefore the use of open data sources and of algorithms and artificial intelligence to gather the data automatically.

Finally—before turning to the English case—the generation and projection of local maps, which make the situation of the districts visible and which can be scaled to the provincial level, seems another relevant line for prevention, which is not excessively complex. The Global Terrorism Database project, to which reference will be made later, is a good example of the usefulness of map projection for the elaboration of diagnoses and rigorous prevention plans.

A forthcoming project is expected to operationalize the index conceptualized briefly so far.

### 4. Starting Hypothesis

This research is based on three assumptions and suggests a hypothesis grounded on several concepts. The first assumption is that it is useful to differentiate between cognitive and behavioral radicalization. Life stories and exhaustive analysis of profiles are required to find common and significant elements, regarding those who choose violence, as the explanatory basis. Factors such as a previous history of violence, an accumulation of minor illegal acts that gradually rupture forms of social control, and recruitment networks of friends, relatives and colleagues would be of significant relevance, as clearly occurs in the Russian case (Lemon et al. 2018) and something less in the subjects that Oliver Roy studied, many of them from France (Roy 2017). Moreover, research of Da Silva on the Portuguese case suggests studying the radicalization process through the analysis of identity positions in former militants' life story narratives, so that understanding the intersection of social, political, historical, and personal circumstances through narrative identity can shed light on how and why certain events take place, and why individuals behave in certain ways (2019).

A second assumption is that thinking and action are connected at some point, so it cannot be suggested that such a strict dissociation between cognitive and behavioral radicalization should be considered. Finding that point is key. What psychologists call strong convictions, awareness-raising (Gardner 1991), may be the link that connects thought and action, especially in the cases where there is no history of previous violence. Once again, a large area of empirical study becomes evident. The third assumption is that, even though violent radicalization should have common elements, when there is a religious motivation, the problem is more complex because motivations are stronger, and the cost–

benefit analyses are no longer so fundamental. The 2012 case, in France, where Merah jumped through a toilet window while shooting as policemen entered to catch him, is indicative of this. Philosophical nihilism does not appear to be motivation which suffices to bring out such behaviors in a relatively large number of people who justify their actions by resorting to divine complacency.

Beyond the explanatory theories, it is known that, as mentioned above, in all those who acted with violence, there was a very strong sense of grievance, they had undergone a major existential crisis promoted by some personal harsh episode, they had a network of contacts linked to violence, and in most cases, they experienced a kind of three-fold uprooting: their family, their society and their religious community.

The crucial concept of the hypothesis consists of connecting radical thinking and violent action—awareness and strong convictions comprise that elusive link—that constitutes the moral structure of behavior. This concept—accompanied by two auxiliary hypotheses: social control, and resilience—could be used to bring together multiple social and psychological nonpathological factors, which make a person choose the path of violence, which currently corresponds to Islamist terrorism.

The moral structure is not a psychological entity (Farid-Arbab 2012; Diessner 2019). It has to do with the internalization of (a) concepts and convictions, (b) thinking, feeling, and behavioral patterns, (c) foreseeing consequences in several courses of action as well as (d) emotional control qualities and competencies, (e) attitudes, (f) motivations that give direction to the purpose, (g) values that determine priorities and (h) a language that has the capacity to connect all those components. Moral structure is the result of a socialization process and, at the same time, also something that the individual can build in a conscious way. Given that this is the key element of our explanation, as we will reveal below, social control also plays a relevant role. For a person to decide to break all the common social conventions in society, and opt for the most extreme way, that of indiscriminate violence, the natural social control mechanisms have had to gradually break down throughout life. This usually takes place in people who have little to lose concerning death, prison, or social margination. It is often said that a very high number of people have suicidal and homicidal thoughts, but very few translate them into reality, depending on the greater or lesser dissolution of bonding to social control (Giner Vidal et al. 2013; Becker [1963] 2009). Finally, individual and collective resilience, related to a strong moral structure, deep scientific-religious knowledge, and strong bonds—friends, family, religious community, and society—would be the main protective elements against external forces.

Moral structure so conceived connects thinking and action but in a very sophisticated way. A person can act with violence because they follow violent behavioral patterns or because they do not have strong enough self-control mechanisms. However, violence for political and religious aims, sustained over time, must be linked to convictions, whether strong or weak. Similarly, an indoctrinated person will only manifest violent behavior if, progressively, he or she is exposed to violence and new patterns related to it, which dissolves little by little his internalized social control mechanisms. It would be very rare for someone with great self-control, peaceful behavior patterns, and strong convictions that delegitimize violence to be indoctrinated and to undergo a change regarding their moral structure. The social group, moreover, is key because moral structure is the result of a group socialization process and the interiorization of norms, so the family, religious and social roots also act as a shield against groups, radicalization agents and group cells which satisfy the desire to belong to those who have said weak bonds.

In this final part, an attempt to put this hypothesis into play will be in line with the data and the several explanations that have been put forward so far to finally approach the case of Spain. Some of the gaps that this hypothesis tries to explain, related to radicalization in different contexts, are the following: some jihadists were so religious while others did not appear to be so; some jihadists had a prior history of violence, while for others this was not the case; some jihadists were poor while others were rich—or, at least, they come from the middle class; some jihadists had higher education, while others did not have

much education; many people legitimize the use of violence and are Salafists, but just a few of them move on to action; some only propagate ideology while others join the armed struggle in Syria or other conflict areas; some deal with recruitment while others engage in attacks; some lead while others commit self-immolation.

Both the rational and the emotional weigh on the moral structure, so, naturally, that the decision to act can be influenced by both normative conviction, strategic conviction, and emotions. In addition, religion, regardless of the degree of knowledge of it, is a very powerful motivational force. For this reason, religion should always be used as one of the explanatory factors, although in dialogue with others. The highest degree of religious knowledge does not, however, always act as protection. If there are strong convictions that delegitimize violence, these will constitute a protective factor; if there are strong convictions that legitimate it, they will be an incentive; and if there are hardly any convictions, the person can be more easily manipulated in one sense or another. The fact that only a percentage of people who justify violence take the final step towards it does not mean from this perspective that thought, and action are separated, but rather that they interact in a sophisticated and dynamic way. Those who have peaceful patterns, roots, and internalized social control mechanisms as well as self-control will need much stronger convictions, more time and more progressive contact with other radicalized environments to end up making the leap to armed struggle. Those who, however, come from a culture marked by violence and criminality, will not need more than an ideology to channel this impulse they have naturalized, although ideology will continue to weigh. The latter phenomenon would be the one that has gained popularity. Oliver Roy calls it Islamizing radicalization, but it is only a variant of a broader phenomenon that takes other forms and assumes different modalities.

The case of Spain until now is somewhat easier because the profile is not as diverse as in other countries. The latest book by the Real Instituto Elcano, already referred to above, *Jihadism and jihadists in Spain. 15 years after 1 M*, is probably the best equipped empirical evidence with explanatory rigor among those published to date, due to the multiple agreements that this think tank has with diverse institutions. Not only does it describe profiling evolution over the last 15 years, but it also identifies what may have been the key factors of radicalization in the case of Spain. It dodges the generalized theoretical explanation, probably because of the somewhat positivistic philosophy of science to which it seems to adhere, but does not in any way reduce the validity of a solid work with great methodological rigor.

There is no need to elaborate on the conclusions here because it has been done in other previously mentioned work (García-Magariño 2018, 2019b), but a summary may be illustrative. From among the more than 200 arrested or killed in the period from 2001 to 2018, a clear profile emerges, although there are differences when comparing the periods before and after 2011, when the war which began in Syria attracted the European population. Most of the arrested or killed terrorists are married men; they are Moroccans or nationalized Spanish citizens with the mentioned origin; the most of them have children—which contrasts with other criminology studies which consider family as a preventive factor of crime (Arias Gallegos 2017); they live in Catalonia, Madrid, Ceuta—and to a lesser degree in Andalusia, Melilla and Valencia; they mostly radicalized during early youth (almost adolescent), in the company of others (90%), in Spain, through a radicalization agent, in places of worship or private homes; they have no criminal record (only 25%), although have previously radicalized relatives or friends; in general terms, they are immigrants, second generations and some converts (10%) with low knowledge of Islam (80%); they are young people (18–35 on average) with secondary or higher education (although slightly lower than the Spanish average); they radicalized in Catalonia, Madrid or Ceuta and in particular in five or six specific municipalities (Madrid, Ceuta, Melilla, Ripoll, Terrassa, Barcelona).

If we divide the periods, we can observe some trends: more women, converts, more youths, Ceuta, Melilla and Catalonia as prominent places of radicalization, emergence of prisons as places of radicalization, more people sentenced for involvement in logistics

related tasks, travel and propaganda (as in the case of women who have not executed any attack in Spain), etc. Regarding the explanatory factors, the Real Instituto Elcano highlights two of them: the agent of on-site radicalization that exposes a person directly to violent ideologies as well as strong social and family bonds regarding previously radicalized people.

Here, again, we suggest that the understanding of this polyhedral reality would be greatly enriched and strengthened, above all, by recourse to the notion of moral structure, and the complementary support provided by social control and resilience as auxiliary explanatory hypotheses. Young Muslims from immigrant families seem to be more vulnerable because, in line with the components of their moral structure, they might be characterized by a lack of strong convictions or be in connection with identity circles and social groups in which there is a certain degree of uprooting and erosion of social control mechanisms due to the lack of recognition and of economic and professional success. They do not have much to lose, but they do have much to gain if they consider the Salafist–jihadist ideology that is presented to them not only in the form of a story full of deep meaning, but also as an identity response, of true friendships, of a sense of mission, of an alternative to injustice, of an explanation to marginality and, in addition, of transcendent ultramundane reward (Silva 2019).

Prevention policies, following this logic, would therefore have to respond to each and every one of the factors identified, if they are to be effective; however, above all, it would seem appropriate to shift the focus from vulnerability to resilience. This, however, will be addressed at a later stage of the ongoing research. Furthermore, the explanatory model also needs to be applied to other countries in Europe and other regions of the world—starting with those with greater cultural closeness, as well as to other forms of radicalization.

## 5. Conclusions

Understanding the causes of jihadist radicalization is as important as it is complex. However, the definition of effective short, medium and long-term policies depends on the understanding of this phenomenon with which we will unfortunately have to coexist for years. In this article (inserted into a larger research project), after developing a small radicalization phenomenology using three main vectors (positive–negative/individualistic–collective/thematic), some of the most consolidated scientific explanatory theories were discussed.

By noting the existing gap in all of them, which comes from the apparent disconnection between radical thinking and violent action, as well as the empirical anomalies that accompany it (they cannot stand the weight of empirical evidence as more data are accumulated), an alternative explanatory hypothesis has been proposed that seeks to fill, although tentatively, this gap, and which is underpinned by the notions of (a) moral structure and (b) the importance of community as the environnment where the individual develops a sense of mission and identity.

After defining the moral structure and describing their components, attempts have been made to discuss its explanatory potential to provide some meaning regarding the existing gap in prevalent explanatory theories. Even on a provisional basis, it could be said that the concept of moral structure seems to have an important heuristic capability to continue studies about violent radicalization, studies that, as has already been mentioned, condition the effectiveness of the response, either as to prevention, direct combat, or de-radicalization.

Besides the conceptual clarification, the theoretical discussion and proposal and the management of certain empirical data coming from jihadism in Spain, this paper delved into two promising methodologies that need to be refined in order to effectively approach radicalization prevention (and probably deradicalization, too): multifactorial social indicators of risk fed by data coming from open sources and correlating the social economic structure of neighborhoods with hate discourses and terrorist attacks.

In conclusion, this work does not come to definitive results, but it seems to set sound basis for a further exploration of four promising areas that are relevant for policies related

to radicalization prevention: conceptual clarity, rigorous theoretical explanation—with heuristic explanatory power—empirical data and appropriate methodological tools.

**Author Contributions:** S.G.M.'s contribution has been to conceptualize the paper, to draft the first version, to propose the methodology, to revise the paper and to edit it. M.J.-R.'s contribution has been to improve the methodology, to do the analysis, to review and improve the first draft and to edit the paper. All authors have read and agreed to the published version of the manuscript.

**Funding:** The paper is within the framework of the Spanish National Project financed by the Ministry of Innovation "Smart War".

**Institutional Review Board Statement:** Not applicable.

**Informed Consent Statement:** Not applicable.

**Data Availability Statement:** Not applicable.

**Conflicts of Interest:** The authors declare no conflict of interests.

## Notes

[1] The rights of this chart belong to the authors of the ABC model. The image have been reproduced from open sources where the chart is explained.

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
