# Peer review of "An Attempt at a Theoretical Explanation of Violent Islamist Radicalization in Spain"

_religions, doi:10.3390/rel13030209_

Round 1
Reviewer 1 Report
This resubmission is stronger and deserves to be published. The one point that needs addressing is a clear statement about the 'aims' of the paper in the Introduction - e.g. the development of a methodology for the empirical study of Islamic radicalisation.
Author Response
- The introduction was enriched by a definition of the main goal of the paper.
- Some spelling mistakes were corrected
- The discussion and conclusions were widened.
- All changes can be seen in the main document as I used the option “track changes”.
- I adjusted the bibliography as there were some inconsistencies. I also added various references.
Reviewer 2 Report
The problem posed in the paper is discussed from an interesting angle. External pull factors and internal push factors are well represented. Preferably, the findings should have been more widely presented. Preferably, the conclusions were presented more widely.
Author Response
- The introduction was enriched by a definition of the main goal of the paper.
- Some spelling mistakes were corrected
- The discussion and conclusions were widened.
- All changes can be seen in the main document as I used the option “track changes”.
- I adjusted the bibliography as there were some inconsistencies. I also added various references.
This manuscript is a resubmission of an earlier submission. The following is a list of the peer review reports and author responses from that submission.
Round 1
Reviewer 1 Report
The main thrust of this paper is to review theories of religious radicalisation. While the review provides a quite comprehensive view of the literature on radicalisation it does not draw clear conclusions about strengths and weaknesses. While the paper states it is looking at Islamic radicalisation in Spain it provides very little data or analysis. It argues for the need to differentiate between cognitive and behavioural radicalisation in order to identify the point at which thinking and action connect to produce violent acts. It offers ‘moral structure’ as a psychological concept to analyse the radicalisation experience – essentially socialisation and international. Yet, the paper only explores these processes abstractly and speculatively. It does not analyse any data about the convergence that leads to violent actions. It just refers the reader to an earlier publication. It refers to the characteristics of 200 jihadists arrested or killed in Spain but not whether their actions were directed against Spanish society or regimes abroad. It notes the profile changed after 2011 but not how.
Recommendations:
- Review theoretical section with the aim of making the argument clearer with a succinct summary
- Provide a more comprehensive description and analysis of the Spanish radicalisation case.
- Undertake a thorough editing of the paper for expression and grammar.
Author Response
Dear reviewer, I am most grafeful for your comments.
I revisited the paper and included both a better theoretical analysis (that is what I hope) and added a empirical case connected with a line of exploration I am undertaking with people of Kent Uni.
The language is also edited.
Thank you for your support and generosity.
Best

Reviewer 2 Report
In its content this brief paper has much to recommend it. It provides a reasonably accurate and helpful survey of the broad development of theories of radicalization and addresses some of the key problems in the literature quite well. It advances several points, that, well not fully original, are still novel, and warrant wider consideration, such as: recognizing that not all radicalization is socially negative, and the mechanisms of radicalization are much the same for positive and negative forms of radicalization up to a point; even many who radicalize in negative (anti-social) ways may not act violently; the key problem, then is discerning what moves people to act violently on their beliefs (cognitions); the delineation of the idea of "moral structure" - and its complex elements - as a way of conceptualizing how and why some people turn from radical attitudes to violent behavior; how this approach incorporates the effects of both rational and emotional factors in creating the conviction that legitimates the turn to violence; and lastly, recognizing the significant and somewhat independent role of religiosity in this process and how it escalates the potential for violence, while also recognizing that under certain conditions a well-develop religious commitment can and usually does protect individuals from becoming violent extremists. All these points are developed in ways that suggest the author understands the more subtle aspects of the debates he is addressing. This article covers a great deal of material in a fairly condensed way and displays a very good knowledge of key issues in the field of radicalization studies (including for example a running critique of Oliver Roy's influential yet suspect theorizing). The attempt to anchor the theorizing in data on jihadist terrorists in Spain in the end is illustrative, but too cursory to demonstrate the validity of the approach. Doing so would require, as the author displays at points they are aware, undertaking a much more careful and detailed analysis of individual cases - that are representative - and that involves actually talking to the terrorists. All of the potential virtues of this paper are undermined, however, by the flawed English, which renders the paper too unclear at important junctures. With my background knowledge I could understand what the author was seeking to say in most cases - but I was rewriting the paper as I read it and interpreting what he probably meant. Finally, a few references are missing from the bibliography.
Author Response
Dear reviewer,
Thank you for the time you took to read the paper and make comments. Your thoughts, in addition, are most welcome.
I attach a new and refined version of the paper that includes a better theoretical exploration and some cases which I think enrich the previous version.
The text is also refined.
All the best

Round 2
Reviewer 1 Report
The revision has improved the data presented to support the overall argument. However, the article requires extensive editing for errors in syntax, grammar and incorrect word choice.
Author Response
Thank you, please see the attachment.

Reviewer 2 Report
While I still think the overall argument, approach, and findings of this manuscript have merit, the language, organization, and presentation of this material is still seriously flawed. The article remains below the acceptable minimum standard for publication - even thought the author(s) had the opportunity to thoroughly revise the paper. Positive changes in the paper were noted, but new material (e.g., about indicators of radicalization and means of countering violent extremism - p. 8) is introduced, for example, without being adequately integrated with the text and logic of the paper. I also found five references missing from the bibliography in just the first eight pages. Overall, though, it remains difficult to assess the precise statements of the author(s) because much of the writing continues to suffer from significant problems related, I suspect, to a lack of facility with English. Such problems exist in almost every paragraph of the text. I detected dozens of such errors in the first three paragraphs of the paper, for example, including a few sentences that simply make no sense as currently written. I have no facility to write well in another language, so I appreciate the talent required to even make this attempt, but this is an English language journal and if this paper were published in its current form, it would be outrageous and harm the reputation of the journal. The author(s) do not seem to have an adequate understanding of the problems and if they wish to persist they need to seek professional editorial assistance.
Author Response
Thank you, please see the attachment.
